# Effects of Moisture Content and Silage Starter on the Fermentation Quality and In Vitro Digestibility of Waxy Corn Processing Byproduct Silage

**Qixuan Yi** [†] , **Meng Yu** [†] , **Peng Wang** *, **Jiarui Du, Tianyue Zhao, Yitong Jin, Hongyu Tang and Bao Yuan** *

College of Animal Sciences, Jilin University, Changchun 130062, China; 13540649853@163.com (Q.Y.); yumeng10165865@163.com (M.Y.); dujiarui626@163.com (J.D.); zty521630@163.com (T.Z.); jinyitong0102@163.com (Y.J.); tanghy@jlu.edu.cn (H.T.)
* Correspondence: pengwang@jlu.edu.cn (P.W.); yuan_bao@jlu.edu.cn (B.Y.)
† These authors contributed equally to this work.

**Abstract:** We investigated the effects of the moisture content and silage starter preparation on the fermentation quality, nutritional value, and in vitro digestibility of waxy corn processing byproducts and rice bran (WRB) mixed silage and waxy corn processing byproducts and rice polished powder (WRPP) mixed silage. Two mixed silages with 55%, 60%, and 65% moisture content (MC) were set up without any additives (control) or with former Lactobacillus (L), and opened on the 60th day after storage the fermentation quality, nutritive value, and in vitro digestibility of the silages in each treatment. The optimal formulation of high-quality waxy corn processing byproduct (WCPP) silage was screened to provide a reference and theoretical basis for the further development and utilisation of WCPPs. The results showed that the proportions of ammonia nitrogen to total nitrogen (AN/TN) and acid detergent lignin (ADL) significantly decreased with a decreasing MC, whereas the levels of lactic acid (LA), crude protein (CP), dry matter (DM), and in vitro crude protein digestibility (IVCPD) significantly increased ($p < 0.05$) for both mixed silages with L. After treatment with 60% MC, the content of neutral detergent fibre (NDF) was significantly lower, and the CP content was significantly greater in the WRB mixed silage treated with L ($p < 0.05$). With 55% MC, the addition of L not only reduced the pH and AN/TN ratio of the two mixed silages but also significantly improved their in vitro digestibility ($p < 0.05$). Studies have shown that reducing the MC of silage raw materials and adding L allows for the preparation of high-quality silage.

**Keywords:** fermentation quality; in vitro digestibility; moisture content; nutritive value; silage starter; waxy corn processing byproducts

## 1. Introduction

In recent years, China's livestock industry has undergone rapid development. At present, the shortage of feed materials is an important factor restricting the development of China's animal husbandry industry. Therefore, to alleviate the shortage of forage feed in the livestock industry, the development of other efficient raw materials as roughage resources has become the main trend in feed development worldwide. China is the world's second largest producer of waxy corn, with 272 million tons produced in 2022 (2022 National Bureau of Statistics of the People's Republic of China). Waxy corn is not only a major food crop but also a major energy ingredient in livestock and poultry feed [1]. A large amount of waxy corn is further processed in China every year to obtain products such as corn starch, ethanol, and monosodium glutamate. The deep processing of waxy corn produces a large amount of waxy corn processing byproducts (WCPPs), which are mostly maize hulls, maize germ meal, maize syrup, and maize alcohol lees. However, most of these WCPPs are incinerated or discarded as agricultural waste without further transformation or comprehensive exploitation. This not only wastes a large amount of potential feed

resources but also brings many problems in terms of the protection and management of the ecological environment.

WCPPs contain a variety of nutrients, such as starch, lipids, proteins, carbohydrates, vitamins, minerals, and trace elements. In recent years, fermented WCPPs have been gradually applied in livestock and poultry production. He et al. [2] used fermented corn germ meal to replace 0, 10%, 20%, and 30% of soybean meal when feeding growing pigs; the results showed that fermented corn germ meal was most effective at replacing 11.80% of the soybean meal and that the average daily intake and average daily weight gain of the growing pigs in the experimental group were greater than those in the control group. According to Wiseman et al. [3], the addition of fermented corn alcohol to liquid diets improves the performance of weaned piglets. The partial replacement of soybean meal with fermented maize alcoholic lees in broiler diets positively affects the feed conversion ratio [4]. Fresh WCPPs are not suitable for prolonged storage, and conventional silage is difficult to use due to the low content of water-soluble carbohydrates (WSCs), low number of attached former Lactobacilli, high moisture content (MC), and high buffering capacity of these plants. Therefore, cofactors with a high dry matter content can be added to the mixture when preparing WCPP silage.

Mixed silage not only reduces the energy loss of the feed but also preserves the quality of the fresh feed and enhances the palatability of the feed. It was found that mixing silage with 40% potato processing byproducts and rice straw reduced the feed-to-weight ratio and enhanced the apparent digestibility. MC and WSCs are key factors in the success of silage [5]. Too much MC will lead to nutrient loss, while too few WSCs will result in the former Lactobacillus not having enough energy to maintain their normal physiological activities during fermentation. Rice bran (RB) and rice polished powder (RPP) are both agricultural byproducts, where RB is a mixture of the pericarp layer, seed coat layer, and germ removed during the finishing process of brown rice, while RPP is a product of paddy processing and has a high protein content. RB is often added as a moisture regulator during silage fermentation to reduce the MC and improve the fermentation quality [6]. When RB and RPP are mixed with WCPPs for silage, the MC of the silage material is regulated, the nutritional balance of the mixed silage material is increased, and the utilisation of agricultural byproducts is improved by increasing the WSC content. Forage mixed with raw materials with a higher dry matter content, such as bran, rice straw, and crop residues, can increase the WSC content and decrease the MC of silage materials [7].

Currently, the exogenous addition of former Lactobacillus preparations has become one of the methods used to enhance the silage quality [8]. Former Lactobacillus, as a type of biological silage starter, can increase the nutritional value of silage while improving silage success, and it is less burdensome and less destructive to the environment. Mugabe et al. [9] reported that the use of *Lactiplantibacillus plantarum* as a silage starter was effective at enhancing the quality of Napier grass silage, accelerating the process of lactic acid fermentation, and reducing the pH and ammonia nitrogen content. Guo et al. [10] reported that the addition of *Lactiplantibacillus plantarum* significantly reduced the pH of *Medicago sativa* silage and promoted lactic acid fermentation, thereby preserving silage nutrients.

However, few studies have reported the addition of former Lactobacillus preparations to silage mixed with WCPPs and auxiliary ingredients. Based on the results of Sudan grass silage [11], we hypothesised that the addition of former Lactobacillus preparations would improve the fermentation quality, nutritive value, and in vitro digestibility of waxy corn processing byproducts and rice bran (WRB) mixed silage and waxy corn processing byproducts and rice polished powder (WRPP) mixed silage under low-MC conditions. Therefore, in this study, RB and RPP were tested for their ability to regulate the MC of WCPP silage materials in different proportions to investigate the effects of the moisture content and silage starter on the fermentation quality, nutritional value, and in vitro digestibility of WRB mixed silage and WRPP mixed silage.

## 2. Materials and Methods

### 2.1. Experimental Materials and Design

The experimental site was located in Jiutai District, Changchun City, Jilin Province (125°24′ to 126°29′ E, 43°50′ to 44°31′ N, 183 m above sea level). The waxy corn processing byproducts (WCPPs) were obtained from Fengze Agriculture Development Co., Ltd. (Changchun, China). The rice bran (RB) and rice polished powder (RPP) were obtained from Jugu Agro Products Ltd. (Bengbu, China). The former Lactobacillus strains (Chikusou-1) were obtained from Snow Brand Seed Co., Ltd., Sapporo, Japan, with the following sequence number—*Lactiplantibacillus plantarum* LP1—and a viable bacterial count of $5 \times 10^7$ CFUs $g^{-1}$ FW (fresh weight).

The RB, RPP, and WCPP were mixed in different proportions, and the theoretical moisture content (MC) of the silage feedstock after mixing was 55%, 60%, and 65%, respectively. The mixing ratios of the WCPP and RB feedstocks were calculated to be 68%:32% (FW), 76%:24% (FW), and 84%:16% (FW), and the mixing ratios of the WCPP and RPP feedstocks were calculated to be 69%:31% (FW), 76%:24% (FW), and 84%:16% (FW), respectively. Two mixed silages (55%, 60%, and 65%) were added to the control and former Lactobacillus (L) groups. The silage starter was immediately sprayed as evenly as possible into the mixed silage using a normal sprayer, with the same dose of distilled water added as a control. After mixing well, the bagged silage method was used; each bag was filled with 500 g of raw material and vacuum-sealed, and six replicates were set up for each treatment. The WRB mixed silage feed and WRPP mixed silage feed were prepared in the dark and stored at room temperature for 60 days.

### 2.2. Fermentation Quality Analysis

The silage bags were opened after 60 days, samples were taken from each of the different treatments to determine the pH, and the ammonia nitrogen (AN) levels were determined using the phenol–sodium hypochlorite colorimetric method [12]. The contents of lactic acid (LA), acetic acid (AA), propionic acid (PA), and butyric acid (BA) were determined using high-performance liquid chromatography (1260 HPLC, Agilent Technologies, Santa Clara, USA) [13].

### 2.3. Chemical Composition and Energy Analysis

In accordance with the methods of the Association of Official Analytical Chemists [14], the dry matter (DM) was determined using the drying method; the crude protein (CP) content was determined using an automatic Kjeldahl nitrogen tester (NKY6160, Wanghai Environmental Technology Co., Ltd., Shanghai, China); the organic matter (OM) and crude ash (Ash) contents were determined using a muffle furnace (MF-N, Dutt Scientific Instruments Ltd., Shanghai, China); the ether extract (EE) content using a Soxhlet extractor (CY-SXT-02, Chuanyi Experimental Instrument Co., Ltd., Shanghai, China); and gross energy (GE) using a fully automatic oxygen bomb calorimeter (ZDHW-8, Brilliance Electronic Technology Co., Ltd., Shijiazhuang, China). The soluble carbohydrate (WSC) content was determined using the anthrone–sulfuric acid colorimetric method [15]. The buffering capacity (BC) was determined via acid–base titration [16]. A fibre analyser (TY-SF22, Tianyan Instrument Co., Ltd., Weifang, China) was used to determine the crude fibre (CF), neutral detergent fibre (NDF), acid detergent fibre (ADF), and acid detergent lignin (ADL) contents [17]. The energy for each category is calculated according to the following formula [18]:

$$DE = GE \times [114.82 - 1.364 \times ADF + 0.104 \times CP + 0.149 \times EE + 0.022 \times NDF - 0.244 \times Ash] / 100 \quad (1)$$

$$ME = 0.82 \times DE \quad (2)$$

$$NEm = 1.37 \times ME - 0.138 \times ME^2 + 0.105 \times ME^3 - 1.12 \quad (3)$$

$$NEf = 1.42 \times ME - 0.174 \times ME^2 + 0.012 \times ME^3 - 1.65 \quad (4)$$

$$NEl = [0.927 - (0.008 \times ADF)]/0.454 \tag{5}$$

Note: DE, digestive energy; ME, metabolic energy; NEm, net energy for maintenance; NEf, net energy for weight gain; NEl, net energy for a lactating cow.

### 2.4. In Vitro Fermentation Parameter Analysis

China Laboratory Animal Welfare Ethics License no. SY202009600. The rumen fluid was obtained from five depopulated small-tailed sheep (mean weight 37 kg) of similar body condition fitted with permanent rumen fistulas. The animals were fed water ad libitum in the morning and evening. The rumen contents of each of the five sheep were collected via a rumen fistula using a vacuum pump, mixed in equal volumes, filtered through four layers of gauze, and immediately added to a glass jar containing buffer (configured according to the method of Longland et al. [19]) to formulate an in vitro culture solution (rumen-fluid-to-buffer ratio of 1:1), which was kept anaerobic by passing through $CO_2$ and kept for use. A total of 0.5 g of the sample was accurately weighed into a filter bag, which was sealed with a plastic sealer and placed into a culture tube with a volume of 100 mL. $CO_2$ was added to the culture tube, and 70 mL of in vitro culture solution was added to the tube, which was quickly capped with a rubber cap and incubated in a 39 °C constant-temperature artificial rumen incubator (BZ-SHH-W21, Biaozhuo Scientific Instruments Co., Ltd., Shanghai, China) for 72 h. At the end of the experiment, the filter bag was removed, quickly rinsed with cold water, and subsequently dried at 65 °C to a constant weight, after which the DM, OM, CP, and NDF residues were removed to calculate the in vitro digestibility [20,21].

### 2.5. Statistical Analysis

The initial collation of the data was performed using THE Microsoft Excel 2010 software, followed by two-way ANOVA using THE SPSS 26.0 software. Tukey's multiple test was used to compare the differences between the groups, with $p < 0.05$ indicating a significant difference.

## 3. Results

The chemical composition, buffering capacity, and energy of the raw materials are shown in Table 1. The DM content of the WCPPs was 25% FW. The CP contents of THE RB and RPP were 6.85% and 5.63% greater than that of THE WCPPS, respectively. The WSC contents in the WCPP, RB, and RPP treatments were 6.06% DM, 9.31% DM, and 8.35% DM, respectively. The BC concentrations were 191.17 mEq kg$^{-1}$ DM for the WCPPs, 176.71 mEq kg$^{-1}$ DM for the RB, and 287.47 mEq kg$^{-1}$ DM for the RPP.

**Table 1.** Chemical composition, buffering capacity, and energy of raw materials.

| Item | WCPP | RB | RPP |
|---|---|---|---|
| Chemical composition and buffering capacity | | | |
| Dry matter (% FW) | 24.55 | 89.22 | 90.12 |
| Organic matter (% DM) | 97.52 | 93.97 | 92.86 |
| Crude protein (% DM) | 11.16 | 18.01 | 16.79 |
| Neutral detergent fibre (% DM) | 75.76 | 58.19 | 69.07 |
| Acid detergent fibre (% DM) | 28.26 | 11.93 | 22.15 |
| Acid detergent lignin (% DM) | 4.50 | 5.91 | 8.63 |
| Water-soluble carbohydrate (% DM) | 6.06 | 9.31 | 8.35 |
| Buffering capacity (mEq kg$^{-1}$ DM) | 191.17 | 176.71 | 287.47 |
| Energy | | | |
| GE (MJ kg$^{-1}$ DM) | 19.98 | 18.74 | 18.50 |
| DE (MJ kg$^{-1}$ DM) | 15.12 | 18.22 | 15.35 |
| ME (MJ kg$^{-1}$ DM) | 12.20 | 14.74 | 12.26 |

**Table 1.** *Cont.*

| Item | WCPP | RB | RPP |
|---|---|---|---|
| NEm (MJ kg$^{-1}$ DM) | 8.89 | 11.49 | 9.13 |
| NEl (MJ kg$^{-1}$ DM) | 7.44 | 9.61 | 7.63 |
| NEf (MJ kg$^{-1}$ DM) | 5.88 | 9.13 | 6.41 |

WCPP, waxy corn processing byproducts; RB, rice bran; RPP, rice polished powder. The same below.

In general, the WSC content of the silage raw materials should exceed 6% DM, and an MC of approximately 60% should be used to make high-quality silage. In the present study, the WSC content in the WCPPs was 6.06% DM, while the MC was as high as 75%. Therefore, it was necessary to manually prepare high-quality silage.

### 3.1. Fermentation Quality of WRB Mixed Silage and WRPP Mixed Silage

The fermentation quality of the WRB mixed silage and WRPP mixed silage are shown in Table 2. For the WRB mixed silage, the AN/TN and BA decreased significantly with a decreasing MC, while the LA content increased significantly in the same L group ($p < 0.05$). After adding L to the 55% MC group, the AN/TN significantly decreased, and the LA content significantly increased ($p < 0.05$). For the WRPP mixed silage, compared with those of the 65% MC group, the pH and AN/TN ratio of the 55% MC group were significantly lower, while the LA content was significantly greater ($p < 0.05$). The addition of L significantly decreased the pH and significantly increased the LA content ($p < 0.05$) at 55% MC.

**Table 2.** Fermentation quality of WRB mixed silage and WRPP mixed silage.

| Item | Silage § | Group | Moisture | | | SEM | *p*-Value | Significance of Main Effects and Interactions | | |
|---|---|---|---|---|---|---|---|---|---|---|
| | | | 55% | 60% | 65% | | | M | G | M × G |
| pH value | WRB | Control | 3.70 | 3.76 | 3.81 | 0.093 | 0.531 | 0.064 | 0.886 | 0.796 |
| | | L | 3.65 | 3.77 | 3.83 | 0.061 | 0.064 | | | |
| | | SEM | 0.122 | 0.006 | 0.059 | | | | | |
| | | *p*-value | 0.703 | 0.158 | 0.753 | | | | | |
| | WRPP | Control | 3.68 [c] | 3.73 [b] | 3.79 [a] | 0.005 | <0.001 | <0.001 | <0.001 | 0.581 |
| | | L | 3.60 [b] | 3.65 [ab] | 3.74 [a] | 0.034 | 0.017 | | | |
| | | SEM | 0.012 | 0.030 | 0.027 | | | | | |
| | | *p*-value | 0.002 | 0.117 | 0.139 | | | | | |
| AN (%TN) | WRB | Control | 2.43 [c] | 2.89 [b] | 3.95 [a] | 0.038 | <0.001 | <0.001 | <0.001 | 0.003 |
| | | L | 2.25 [c] | 2.55 [b] | 3.47 [a] | 0.059 | <0.001 | | | |
| | | SEM | 0.040 | 0.046 | 0.061 | | | | | |
| | | *p*-value | 0.012 | 0.002 | 0.001 | | | | | |
| | WRPP | Control | 2.40 [c] | 2.57 [b] | 3.36 [a] | 0.044 | <0.001 | <0.001 | 0.001 | 0.517 |
| | | L | 2.17 [b] | 2.38 [b] | 3.00 [a] | 0.144 | 0.003 | | | |
| | | SEM | 0.175 | 0.034 | 0.047 | | | | | |
| | | *p*-value | 0.259 | 0.023 | 0.002 | | | | | |
| LA (%DM) | WRB | Control | 8.89 | 10.11 | 7.90 | 0.949 | 0.144 | <0.001 | 0.762 | 0.003 |
| | | L | 12.37 [a] | 9.57 [b] | 5.47 [c] | 0.952 | 0.001 | | | |
| | | SEM | 0.804 | 0.820 | 1.180 | | | | | |
| | | *p*-value | 0.012 | 0.544 | 0.109 | | | | | |
| | WRPP | Control | 6.03 [c] | 10.44 [a] | 8.23 [b] | 0.507 | <0.001 | 0.036 | 0.038 | <0.001 |
| | | L | 11.27 [a] | 8.45 [b] | 7.76 [b] | 0.831 | 0.012 | | | |
| | | SEM | 0.974 | 0.312 | 0.613 | | | | | |
| | | *p*-value | 0.006 | 0.003 | 0.483 | | | | | |

**Table 2.** *Cont.*

| Item | Silage [§] | Group | Moisture | | | SEM | *p*-Value | Significance of Main Effects and Interactions | | |
|---|---|---|---|---|---|---|---|---|---|---|
| | | | 55% | 60% | 65% | | | M | G | M × G |
| AA (%DM) | WRB | Control | 2.05 | 2.85 | 2.58 | 0.421 | 0.233 | 0.238 | 0.706 | 0.059 |
| | | L | 2.83 | 2.59 | 1.78 | 0.425 | 0.105 | | | |
| | | SEM | 0.555 | 0.208 | 0.430 | | | | | |
| | | *p*-value | 0.233 | 0.285 | 0.134 | | | | | |
| | WRPP | Control | 2.02 | 2.23 | 2.34 | 0.382 | 0.719 | 0.911 | 0.719 | 0.740 |
| | | L | 2.14 | 2.19 | 1.99 | 0.462 | 0.905 | | | |
| | | SEM | 0.628 | 0.342 | 0.166 | | | | | |
| | | *p*-value | 0.862 | 0.913 | 0.105 | | | | | |
| PA (%DM) | WRB | Control | ND | ND | ND | NA | NA | NA | NA | NA |
| | | L | ND | ND | ND | NA | NA | | | |
| | | SEM | NA | NA | NA | | | | | |
| | | *p*-value | NA | NA | NA | | | | | |
| | WRPP | Control | 0.02 | 0.01 | ND | 0.014 | 0.386 | 0.226 | 0.690 | 0.946 |
| | | L | 0.02 | 0.01 | ND | 0.015 | 0.553 | | | |
| | | SEM | 0.023 | 0.009 | NA | | | | | |
| | | *p*-value | 0.890 | 0.519 | NA | | | | | |
| BA (%DM) | WRB | Control | 0.14 [b] | 0.62 [a] | 0.67 [a] | 0.137 | 0.016 | <0.001 | 0.762 | 0.132 |
| | | L | 0.26 [c] | 0.70 [a] | 0.47 [b] | 0.075 | 0.003 | | | |
| | | SEM | 0.063 | 0.135 | 0.119 | | | | | |
| | | *p*-value | 0.139 | 0.585 | 0.175 | | | | | |
| | WRPP | Control | 0.36 | 0.41 | 0.50 | 0.073 | 0.259 | 0.245 | 0.980 | 0.722 |
| | | L | 0.39 | 0.44 | 0.45 | 0.080 | 0.726 | | | |
| | | SEM | 0.110 | 0.065 | 0.037 | | | | | |
| | | *p*-value | 0.842 | 0.670 | 0.246 | | | | | |

Different lowercase letters in the same row indicate significant differences ($p < 0.05$). SEM, standard error of the mean; ND, not detected; NA, not applicable; M, moisture content; G, group; [§] WRB, waxy corn processing byproduct and rice bran; WRPP, waxy corn processing byproduct and rice polished powder.

For the WRB mixed silage, M, A, and the interaction M × A influenced the proportion of ammonia nitrogen to total nitrogen (AN/TN) ($p = 0.003$; $p < 0.001$); M and the interaction M × A affected the LA content ($p = 0.003$; $p < 0.001$); and M treatment affected the BA content ($p < 0.001$). For the WRPP mixed silage, M, A, and the interaction M × A affected the LA content ($p = 0.036$–$0.038$; $p < 0.001$); the M and A treatments affected the pH and AN/TN ratio ($p \leq 0.001$).

Overall, the best results were obtained with the addition of L to the WRB mixed silage and WRPP mixed silage at 55% MC, which significantly improved the fermentation quality of the silage.

### 3.2. Chemical Composition of WRB Mixed Silage and WRPP Mixed Silage

The chemical compositions of the WRB mixed silage and WRPP mixed silage are shown in Table 3. For both mixed silages, under the same L condition, compared with those in the 65% MC treatment group, the OM and ADL contents in the 55% MC treatment group were significantly lower, while the DM and CP contents were significantly greater ($p < 0.05$). After the addition of L to the 60% MC treatment group, the CP content of the WRB mixed silage significantly increased, while the NDF content significantly decreased ($p < 0.05$). The addition of L significantly increased the OM and CP contents and significantly decreased the ADF content of the WRPP mixed silage at 55% MC ($p < 0.05$).

Table 3. Chemical composition of WRB mixed silage and WRPP mixed silage.

| Item | Silage [§] | Group | Moisture | | | SEM | *p*-Value | Significance of Main Effects and Interactions | | |
|---|---|---|---|---|---|---|---|---|---|---|
| | | | 55% | 60% | 65% | | | M | G | M × G |
| DM (%FW) | WRB | Control | 40.68 [a] | 36.01 [b] | 31.66 [c] | 0.442 | <0.001 | <0.001 | 0.823 | 0.914 |
| | | L | 40.61 [a] | 36.10 [b] | 31.78 [c] | 0.205 | <0.001 | | | |
| | | SEM | 0.448 | 0.358 | 0.162 | | | | | |
| | | *p*-value | 0.878 | 0.807 | 0.511 | | | | | |
| | WRPP | Control | 40.04 [a] | 35.77 [b] | 32.15 [c] | 0.178 | <0.001 | <0.001 | 0.771 | 0.150 |
| | | L | 40.30 [a] | 35.72 [b] | 31.84 [c] | 0.208 | <0.001 | | | |
| | | SEM | 0.196 | 0.229 | 0.147 | | | | | |
| | | *p*-value | 0.245 | 0.817 | 0.103 | | | | | |
| OM (%DM) | WRB | Control | 95.25 [c] | 95.62 [b] | 96.18 [a] | 0.068 | <0.001 | <0.001 | 0.462 | 0.784 |
| | | L | 95.22 [c] | 95.57 [b] | 96.18 [a] | 0.040 | <0.001 | | | |
| | | SEM | 0.089 | 0.020 | 0.031 | | | | | |
| | | *p*-value | 0.727 | 0.083 | 0.842 | | | | | |
| | WRPP | Control | 94.57 [c] | 95.16 [b] | 96.03 [a] | 0.020 | <0.001 | <0.001 | 0.014 | <0.001 |
| | | L | 94.69 [c] | 95.04 [b] | 95.89 [a] | 0.038 | <0.001 | | | |
| | | SEM | 0.037 | 0.023 | 0.031 | | | | | |
| | | *p*-value | 0.035 | 0.005 | 0.009 | | | | | |
| CP (%DM) | WRB | Control | 18.54 [a] | 17.33 [b] | 15.24 [c] | 0.152 | <0.001 | <0.001 | 0.623 | 0.003 |
| | | L | 18.88 [a] | 17.83 [b] | 15.51 [c] | 0.137 | <0.001 | | | |
| | | SEM | 0.131 | 0.123 | 0.175 | | | | | |
| | | *p*-value | 0.059 | 0.017 | 0.193 | | | | | |
| | WRPP | Control | 17.55 [a] | 16.52 [b] | 14.84 [c] | 0.118 | <0.001 | <0.001 | <0.001 | 0.510 |
| | | L | 17.94 [a] | 16.75 [b] | 15.27 [c] | 0.123 | <0.001 | | | |
| | | SEM | 0.111 | 0.147 | 0.100 | | | | | |
| | | *p*-value | 0.025 | 0.182 | 0.013 | | | | | |
| NDF (%DM) | WRB | Control | 63.17 [c] | 67.82 [a] | 64.29 [b] | 0.761 | <0.001 | <0.001 | 0.081 | 0.013 |
| | | L | 61.99 [b] | 65.67 [a] | 65.58 [a] | 0.592 | 0.015 | | | |
| | | SEM | 0.967 | 0.383 | 0.559 | | | | | |
| | | *p*-value | 0.287 | 0.005 | 0.082 | | | | | |
| | WRPP | Control | 61.58 | 62.60 | 62.33 | 0.520 | 0.210 | 0.005 | 0.486 | 0.214 |
| | | L | 60.30 [b] | 62.89 [a] | 62.50 [ab] | 0.776 | 0.032 | | | |
| | | SEM | 1.089 | 0.224 | 0.271 | | | | | |
| | | *p*-value | 0.304 | 0.271 | 0.558 | | | | | |
| ADF (%DM) | WRB | Control | 17.79 [c] | 19.68 [b] | 21.38 [a] | 0.571 | 0.002 | <0.001 | 0.589 | 0.609 |
| | | L | 17.56 [c] | 19.86 [b] | 22.04 [a] | 0.673 | 0.002 | | | |
| | | SEM | 0.374 | 0.673 | 0.759 | | | | | |
| | | *p*-value | 0.562 | 0.806 | 0.433 | | | | | |
| | WRPP | Control | 25.14 | 24.70 | 25.40 | 0.555 | 0.487 | 0.348 | 0.761 | 0.111 |
| | | L | 24.50 [b] | 25.70 [a] | 25.31 [ab] | 0.468 | 0.101 | | | |
| | | SEM | 0.181 | 0.471 | 0.733 | | | | | |
| | | *p*-value | 0.024 | 0.100 | 0.915 | | | | | |
| ADL (%DM) | WRB | Control | 5.51 [c] | 6.26 [a] | 5.84 [b] | 0.083 | <0.001 | 0.001 | 0.573 | 0.099 |
| | | L | 5.47 [b] | 6.07 [ab] | 6.26 [a] | 0.255 | 0.048 | | | |
| | | SEM | 0.083 | 0.308 | 0.078 | | | | | |
| | | *p*-value | 0.671 | 0.570 | 0.006 | | | | | |
| | WRPP | Control | 7.97 | 8.42 | 9.06 | 0.477 | 0.151 | 0.005 | 0.867 | 0.536 |
| | | L | 7.86 [b] | 8.80 [a] | 8.90 [a] | 0.220 | 0.006 | | | |
| | | SEM | 0.581 | 0.198 | 0.192 | | | | | |
| | | *p*-value | 0.858 | 0.124 | 0.461 | | | | | |

Different lowercase letters in the same row indicate significant differences (*p* < 0.05). SEM, standard error of the mean; M, moisture content; G, group; [§] WRB, waxy corn processing byproduct and rice bran; WRPP, waxy corn processing byproduct and rice polished powder.

For the WRB mixed silage, the M treatment affected the DM, OM, ADF, and ADL contents ($p \leq 0.001$); M and the interaction M × A affected the CP and NDF contents ($p = 0.003$–$0.013$; $p < 0.001$). For the WRPP mixed silage, the M treatment affected the DM, NDF, and ADL contents ($p = 0.005$; $p < 0.001$); the M and A treatments affected the CP content ($p < 0.001$); and the M, A, and interaction M × A affected the OM content ($p = 0.014$; $p < 0.001$).

Based on the above results, both MC reduction and L addition are recommended for WRB mixed silage and WRPP mixed silage to improve the nutritional value of the silage.

### 3.3. Energy of WRB Mixed Silage and WRPP Mixed Silage

The energy consumption of the WRB mixed silage and WRPP mixed silage is shown in Table 4. For both mixed silages, the GE, DE, ME, NEm, NEl, and NEf were the highest in the 55% MC treatment group and were significantly different ($p < 0.05$) from those in the 60% MC treatment group. Under 55% MC conditions, compared with those in the control group, the DE, ME, NEm, NEl, and NEf in the L group were significantly greater ($p < 0.05$).

**Table 4.** Energy of WRB mixed silage and WRPP mixed silage.

| Item | Silage [§] | Group | Moisture | | | SEM | *p*-Value | Significance of Main Effects and Interactions | | |
|---|---|---|---|---|---|---|---|---|---|---|
| | | | 55% | 60% | 65% | | | M | G | M × G |
| GE (MJ kg$^{-1}$ DM) | WRB | Control | 19.35 [b] | 19.52 [a] | 19.60 [a] | 0.053 | 0.008 | <0.001 | 0.964 | 0.011 |
| | | L | 19.55 [a] | 19.45 [b] | 19.47 [b] | 0.027 | 0.021 | | | |
| | | SEM | 0.056 | 0.029 | 0.035 | | | | | |
| | | *p*-value | 0.094 | 0.059 | 0.222 | | | | | |
| | WRPP | Control | 20.81 | 20.94 | 20.77 | 0.160 | 0.561 | 0.860 | 0.452 | 0.076 |
| | | L | 21.01 [a] | 20.77 [c] | 20.90 [b] | 0.041 | 0.003 | | | |
| | | SEM | 0.094 | 0.038 | 0.174 | | | | | |
| | | *p*-value | 0.439 | 0.010 | 0.288 | | | | | |
| DE (MJ kg$^{-1}$ DM) | WRB | Control | 17.32 [a] | 16.99 [ab] | 16.58 [b] | 0.184 | 0.020 | <0.001 | 0.636 | 0.270 |
| | | L | 17.52 [a] | 16.87 [b] | 16.35 [c] | 0.189 | 0.002 | | | |
| | | SEM | 0.116 | 0.197 | 0.228 | | | | | |
| | | *p*-value | 0.161 | 0.565 | 0.363 | | | | | |
| | WRPP | Control | 16.68 [ab] | 16.92 [a] | 16.55 [b] | 0.110 | 0.040 | 0.357 | 0.783 | 0.005 |
| | | L | 16.94 [a] | 16.47 [b] | 16.80 [ab] | 0.159 | 0.064 | | | |
| | | SEM | 0.090 | 0.133 | 0.174 | | | | | |
| | | *p*-value | 0.044 | 0.029 | 0.229 | | | | | |
| ME (MJ kg$^{-1}$ DM) | WRB | Control | 14.04 [a] | 13.78 [ab] | 13.49 [b] | 0.142 | 0.024 | <0.001 | 0.590 | 0.232 |
| | | L | 14.20 [a] | 13.69 [b] | 13.28 [c] | 0.150 | 0.003 | | | |
| | | SEM | 0.090 | 0.158 | 0.176 | | | | | |
| | | *p*-value | 0.149 | 0.586 | 0.306 | | | | | |
| | WRPP | Control | 13.42 [ab] | 13.64 [a] | 13.41 [b] | 0.088 | 0.075 | 0.596 | 0.836 | 0.004 |
| | | L | 13.65 [a] | 13.28 [b] | 13.59 [a] | 0.127 | 0.053 | | | |
| | | SEM | 0.073 | 0.107 | 0.137 | | | | | |
| | | *p*-value | 0.036 | 0.029 | 0.275 | | | | | |
| NEm (MJ kg$^{-1}$ DM) | WRB | Control | 10.70 [a] | 10.43 [ab] | 10.14 [b] | 0.132 | 0.015 | <0.001 | 0.590 | 0.267 |
| | | L | 10.84 [a] | 10.35 [b] | 9.95 [c] | 0.139 | 0.002 | | | |
| | | SEM | 0.081 | 0.147 | 0.164 | | | | | |
| | | *p*-value | 0.168 | 0.615 | 0.320 | | | | | |
| | WRPP | Control | 9.92 | 10.10 | 9.91 | 0.080 | 0.091 | 0.588 | 0.909 | 0.006 |
| | | L | 10.12 [a] | 9.79 [b] | 10.05 [ab] | 0.114 | 0.061 | | | |
| | | SEM | 0.058 | 0.099 | 0.126 | | | | | |
| | | *p*-value | 0.026 | 0.034 | 0.351 | | | | | |

**Table 4.** *Cont.*

| Item | Silage [§] | Group | Moisture | | | SEM | *p*-Value | Significance of Main Effects and Interactions | | |
|---|---|---|---|---|---|---|---|---|---|---|
| | | | 55% | 60% | 65% | | | M | G | M × G |
| NEl (MJ kg$^{-1}$ DM) | WRB | Control | 8.94 [a] | 8.72 [ab] | 8.48 [b] | 0.110 | 0.016 | <0.001 | 0.607 | 0.251 |
| | | L | 9.06 [a] | 8.65 [b] | 8.32 [c] | 0.116 | 0.002 | | | |
| | | SEM | 0.070 | 0.123 | 0.135 | | | | | |
| | | *p*-value | 0.161 | 0.617 | 0.311 | | | | | |
| | WRPP | Control | 8.29 | 8.45 | 8.29 | 0.066 | 0.083 | 0.584 | 0.891 | 0.005 |
| | | L | 8.46 [a] | 8.18 [b] | 8.40 [ab] | 0.095 | 0.058 | | | |
| | | SEM | 0.049 | 0.083 | 0.105 | | | | | |
| | | *p*-value | 0.025 | 0.033 | 0.329 | | | | | |
| NEf (MJ kg$^{-1}$ DM) | WRB | Control | 8.03 [a] | 7.67 [b] | 7.32 [c] | 0.143 | 0.007 | <0.001 | 0.602 | 0.328 |
| | | L | 8.16 [a] | 7.60 [b] | 7.12 [c] | 0.159 | 0.002 | | | |
| | | SEM | 0.093 | 0.164 | 0.181 | | | | | |
| | | *p*-value | 0.226 | 0.664 | 0.338 | | | | | |
| | WRPP | Control | 6.84 | 7.01 | 6.83 | 0.096 | 0.180 | 0.572 | 0.930 | 0.013 |
| | | L | 7.04 [a] | 6.70 [b] | 6.93 [ab] | 0.117 | 0.069 | | | |
| | | SEM | 0.052 | 0.108 | 0.142 | | | | | |
| | | *p*-value | 0.018 | 0.044 | 0.532 | | | | | |

Different lowercase letters in the same row indicate significant differences ($p < 0.05$). SEM, standard error of the mean; M, moisture content; G, group; [§] WRB, waxy corn processing byproduct and rice bran; WRPP, waxy corn processing byproduct and rice polished powder.

For the WRB mixed silage, the M treatment affected the DE, ME, NEm, NEl, and NEf ($p < 0.001$), while M and the interaction M×A affected the GE ($p = 0.011$; $p < 0.001$). For the WRPP mixed silage, the interaction M × A affected the DE, ME, NEm, NEl, and NEf ($p = 0.004$–$0.013$).

Overall, the addition of L to the WRB mixed silage and WRPP mixed silage at 55% MC was the most effective and significantly improved the energy of the silage.

*3.4. In Vitro Digestibility of WRB Mixed Silage and WRPP Mixed Silage*

The in vitro digestibility of the WRB mixed silage and WRPP mixed silage is shown in Table 5. For both mixed silages, the IVCPD and IVNDFD increased significantly ($p < 0.05$) with a decreasing MC in the same L group. Under 55% MC conditions, the addition of L significantly increased the IVCPD and IVNDFD of the WRB mixed silage and significantly increased the IVDMD and IVOMD of the WRPP mixed silage ($p < 0.05$).

**Table 5.** In vitro digestibility of WRB mixed silage and WRPP mixed silage.

| Item [‡] | Silage [§] | Group | Moisture | | | SEM | *p*-Value | Significance of Main Effects and Interactions | | |
|---|---|---|---|---|---|---|---|---|---|---|
| | | | 55% | 60% | 65% | | | M | G | M × G |
| IVDMD (%DM) | WRB | Control | 67.59 [a] | 66.55 [b] | 65.12 [c] | 0.376 | 0.002 | <0.001 | 0.490 | 0.416 |
| | | L | 67.87 [a] | 66.21 [b] | 64.70 [c] | 0.408 | <0.001 | | | |
| | | SEM | 0.234 | 0.403 | 0495 | | | | | |
| | | *p*-value | 0.302 | 0.446 | 0.444 | | | | | |
| | WRPP | Control | 64.52 [a] | 64.74 [a] | 63.97 [b] | 0.221 | 0.032 | 0.032 | 0.913 | 0.010 |
| | | L | 64.98 [a] | 63.93 [b] | 64.38 [ab] | 0.319 | 0.045 | | | |
| | | SEM | 0.128 | 0.285 | 0.359 | | | | | |
| | | *p*-value | 0.023 | 0.046 | 0.321 | | | | | |

**Table 5.** *Cont.*

| Item [‡] | Silage [§] | Group | Moisture | | | SEM | *p*-Value | Significance of Main Effects and Interactions | | |
|---|---|---|---|---|---|---|---|---|---|---|
| | | | **55%** | **60%** | **65%** | | | **M** | **G** | **M × G** |
| IVOMD (%DM) | WRB | Control | 72.48 [a] | 71.26 [b] | 69.85 [c] | 0.390 | 0.002 | <0.001 | 0.510 | 0.580 |
| | | L | 72.67 [a] | 71.00 [b] | 69.43 [c] | 0.447 | 0.001 | | | |
| | | SEM | 0.242 | 0.437 | 0.528 | | | | | |
| | | *p*-value | 0.476 | 0.579 | 0.471 | | | | | |
| | WRPP | Control | 67.90 | 68.17 | 67.59 | 0.307 | 0.250 | 0.176 | 0.840 | 0.049 |
| | | L | 68.34 [a] | 67.42 [b] | 67.79 [ab] | 0.327 | 0.078 | | | |
| | | SEM | 0.102 | 0.311 | 0.442 | | | | | |
| | | *p*-value | 0.012 | 0.073 | 0.684 | | | | | |
| IVCPD (%DM) | WRB | Control | 62.80 [a] | 62.02 [b] | 59.29 [c] | 0.196 | <0.001 | <0.001 | 0.097 | 0.022 |
| | | L | 63.37 [a] | 61.67 [b] | 59.73 [c] | 0.232 | <0.001 | | | |
| | | SEM | 0.233 | 0.153 | 0.246 | | | | | |
| | | *p*-value | 0.070 | 0.086 | 0.144 | | | | | |
| | WRPP | Control | 55.08 | 54.45 | 53.95 | 0.546 | 0.196 | 0.001 | 0.467 | 0.238 |
| | | L | 55.63 [a] | 55.01 [a] | 53.45 [b] | 0.401 | 0.004 | | | |
| | | SEM | 0.408 | 0.276 | 0.668 | | | | | |
| | | *p*-value | 0.246 | 0.109 | 0.498 | | | | | |
| IVNDFD (%DM) | WRB | Control | 60.73 [a] | 59.39 [ab] | 58.31 [b] | 0.732 | 0.044 | <0.001 | 0.082 | 0.012 |
| | | L | 62.80 [a] | 60.63 [b] | 57.17 [c] | 0.572 | <0.001 | | | |
| | | SEM | 0.371 | 0.540 | 0.930 | | | | | |
| | | *p*-value | 0.005 | 0.083 | 0.285 | | | | | |
| | WRPP | Control | 57.76 | 57.50 | 56.78 | 0.503 | 0.210 | 0.005 | 0.488 | 0.215 |
| | | L | 58.04 [a] | 57.67 [ab] | 55.54 [b] | 0.748 | 0.032 | | | |
| | | SEM | 0.213 | 0.260 | 1.052 | | | | | |
| | | *p*-value | 0.265 | 0.549 | 0.305 | | | | | |

Different lowercase letters in the same row indicate significant differences ($p < 0.05$). SEM, standard error of the mean; M, moisture content; G, group. [‡] DM: dry matter; IVDMD, in vitro dry matter digestibility; IVOMD, in vitro organic matter digestibility; IVCPD, in vitro crude protein digestibility; IVNDFD, in vitro neutral detergent fibre digestibility; [§] WRB, waxy corn processing byproduct and rice bran; WRPP, waxy corn processing byproduct and rice polished powder.

For the WRB mixed silage, the M treatment the affected IVDMD and IVOMD ($p < 0.001$), while M and the interaction M × A affected the IVCPD and IVNDFD ($p = 0.012$–$0.022$; $p < 0.001$). For the WRPP mixed silage, the M treatment affected the IVCPD and IVNDFD ($p = 0.001$–$0.005$); the M treatment and the interaction M × A affected the IVDMD ($p = 0.010$–$0.032$); and the interaction M × A affected IVOMD ($p = 0.049$).

Based on the above results, both MC reduction and L addition were recommended for WRB mixed silage and WRPP mixed silage to improve the in vitro digestibility of the silage.

## 4. Discussion

### 4.1. Effects of Moisture Content and Silage Starter on the Fermentation Quality of WRB Mixed Silage and WRPP Mixed Silage

In general, the WSC content of the silage ingredients should be greater than 6% DM to ensure good-quality silage, and it is not easy to make good-quality silage when the WSC content is less than 2% DM [22]. According to McDonald [23], pastureland with a raw material MC of up to 80% did not inhibit the reproductive growth of clostridia or mould even when the pH was reduced to 4.0. In this trial, the WCPPs had a WSC content of 6.06% DM, while the MC content was as high as 75%; therefore, the WCPPs would also need to be artificially conditioned to make good-quality silage. A suitable MC concentration not only improves the fermentation quality of silage but also reduces silage permeate production and nutrient loss. A too-high MC in silage material is not favourable for the fermentation of former Lactobacillus in the pre-silage phase, leading to an increase in the volatile fatty acid

content. Furthermore, too high of an MC results in soluble nutrients being excreted with the exudate, affecting the nutritional value of the silage [24]. A too-low MC will increase the difficulty of compacting the silage material and reduce the activity of water in the medium; additionally, the acid-producing bacteria will experience a physiological drought, and the accumulation of acidity in the silage will be inhibited, which is unfavourable for silage fermentation. In this experiment, the WRB mixed silage and WRPP mixed silage from the 55% MC treatment group had the lowest pH and AN/TN and the highest LA content compared to those of the other MC treatment groups. This is explained by the fact that a proper reduction in the water content of the silage material concentrates nutrients such as WSCs, which favour fermentation by former Lactobacillus, reduce the pH, and enhance the silage quality, which is in agreement with the experimental results of Owens et al. [25]. Studies have shown that the use of RB as a carbon source promotes fermentation by former Lactobacillus strains and increases the silage LA content [26]. RB and RPP, as auxiliaries, while regulating the moisture content of silage material, can also enhance the quality of silage by increasing the WSC content to improve the nutritional balance of the mixed silage material and by promoting the growth and multiplication of former Lactobacillus to inhibit undesirable microbial activities in the silage. Wan et al. [11] reported that with the addition of the same *Lactiplantibacillus plantarum* strain, the pH and AN/TN ratio of Sudan grass silage gradually decreased with a decreasing MC, which was consistent with the results of the present study. Azevedo et al. [27] subjected silage material to wilting treatment and reported that the pH and AN/TN ratio of Xaraes and Piata palisade grass silages in the wilted group were significantly lower than those in the control group and that the fermentation quality of Xaraes and Piata palisade grass silages was better, which was consistent with the results of the present study. In this study, the BA content of the WRB mixed silage from the 55% MC treatment was significantly lower than that from the 65% MC treatment, indicating that low MC can cause physiological drying of microorganisms and that the inhibitory effect on clostridia, mould, etc., may be greater than that on Lactobacilli, which reduces the BA content of the silage. The AN content is related to the degree of protein and amino acid decomposition in the silage; the higher the AN/TN ratio is, the greater the protein and amino acid decomposition, and the poorer the silage fermentation quality [28]. The two silages had lower AN/TN ratios under low-MC conditions, indicating that the low-MC treatment slowed the degree of protein degradation in the silage. This may be because as the MC concentration decreases, the plant cytosol becomes thicker, and the osmotic pressure increases, which partially inhibits the activity of undesirable microorganisms, enzyme catabolism, and respiration of the plant cells and reduces the rate of protein hydrolysis; as a result, the fermentation quality of both the WRB mixed silage and the WRPP mixed silage increases [29]. Hristov et al. [30] investigated protein hydrolysis and rumen degradation rates in alfalfa silages with different MCs and showed that low-MC alfalfa silages had lower nonprotein nitrogen and ammonia nitrogen contents than high-MC alfalfa silages and that the rumen protein degradation rates decreased with a decreasing MC. As shown in Table 2, the fermentation quality of the WRB mixed silage and WRPP mixed silage increased with a decreasing raw material MC, indicating that silage at 55% MC is an effective way of storing WCPPs.

There are many microorganisms present in fresh forage, of which the number of harmful microorganisms is much greater than the number of beneficial microorganisms, and harmful microorganisms are detrimental to silage, often resulting in nutrient losses. The microorganisms adhering to the surface of different silage materials also vary considerably in type and number. Former Lactobacillus species are important flora for silage fermentation, and the number of former Lactobacillus species attached to the forage is closely related to the fermentation process. A minimum of $10^5$ CFU $g^{-1}$ of former Lactobacillus in the silage material was needed for a rapid decrease in pH, the inhibition of harmful microbial activities such as clostridia and mould, and a reduction in nutrient consumption [31]. However, the number of former Lactobacillus on the surface of silage material is generally insufficient, so it is necessary to add additional former Lactobacillus to the silage material to ensure

that lactic acid fermentation is dominant during the ensiling process. The important former Lactobacillus in silage can be categorised into two types of bacteria: homofermentative and heterofermentative. Homofermentative former Lactobacillus produce lactic acid from glucose by glycolysis, while heterofermentative former Lactobacillus products include ethanol, acetic acid, and carbon dioxide, in addition to lactic acid. Comparatively speaking, plants subjected to homofermentation are more able to fully utilise nutrients and reduce nutrient losses. Therefore, a homofermentative former Lactobacillus (*Lactiplantibacillus plantarum*) was selected for silage fermentation in this experiment. To improve the characteristics of silage and, consequently, animal nutrition, the use of the former Lactobacillus inoculant should meet the following criteria: 1. The effective concentration that allowed former Lactobacillus to proliferate rapidly and dominate in competition with other microorganisms was $10^5$ CFU g$^{-1}$; 2. A homologous fermentation pathway is available to maximise the production of lactic acid from hexasaccharides; 3. Acid tolerance was defined as a pH reduction below 4.0 as quickly as possible to inhibit the activity of other microorganisms; 4. Survival in low-moisture (wilted silage) feedstocks; 5. Inability to hydrolyse proteins.

pH is an important indicator of silage quality. The pH of good-quality silage should be less than 4.2, and a pH greater than 4.2 is more likely to cause the spoilage of the silage [32]. In this experiment, the pH of the L group was less than 4.2 under different MC conditions, indicating that both the WRB mixed silage and the WRPP mixed silage with L added were high-quality silage. Li et al. [33] showed that the addition of *Lactiplantibacillus plantarum* to the silage decreased the pH and PA content, increased the LA content, and improved the fermentation quality of Stylo (*Stylosanthes guianensis*) silage, which was consistent with the results of the present study. Similarly, Khota et al. [34] showed that, compared with a control, the inoculation of sorghum silage with *Lactiplantibacillus plantarum* resulted in an increased LA content and reduced AN/TN. Zhang et al. [35] added *Lactiplantibacillus plantarum* to alfalfa silage and reported that *Lactiplantibacillus plantarum* improved the fermentation quality of the alfalfa silage by increasing the proportion of former Lactobacillus strains. This is explained by the fact that after the addition of the former Lactobacillus liquid, the former Lactobacillus population reaches a dominant state and multiplies rapidly. This promotes the fermentation process, dominated by lactic acid fermentation at the early stage of silage fermentation, with a large accumulation of LA, which rapidly reduces the pH of silage, inhibits the activity of undesirable bacterial flora, and enhances the quality of silage fermentation [36]. The degradation of CP by plant enzymes and microorganisms produces ammonia nitrogen in silage, and the AN/TN ratio reflects the degree of CP degradation in the silage [28]. Additionally, the pH affects the activity of proteolytic enzymes in silage, and protein degradation and decarboxylation reactions in the silage are weakened by acidification, thus reducing the ammoniacal nitrogen content. The AN/TN ratio of both the WRB mixed silage and the WRPP mixed silage from the L group was significantly lower than that in the control group at 55% MC, suggesting that the L group had better fermentation. This may be because the addition of L creates a microacidic environment, which promotes a rapid decrease in pH, inhibits the activity of aerobic microorganisms and plant enzymes, and decreases the free ammonia content in the fermentation system, which in turn reduces the AN/TN. PA and BA have an unpleasant, irritating odour and are products of protein breakdown by harmful microorganisms such as moulds and clostridia, and their high content seriously affects the palatability of silage, with higher levels of PA and BA indicating poorer silage fermentation quality [37]. In this study, PA was not detected in any of the treatment groups, except for a very low level of PA detected in the WRPP silage mixes at 55% and 60% MC, which also suggested that the addition of former Lactobacillus promoted lactic acid fermentation, reduced the production of harmful secondary metabolites, and enhanced silage quality. Therefore, the addition of *Lactiplantibacillus plantarum* positively affected the fermentation quality of the WRB mixed silage and WRPP mixed silage.

*4.2. Effects of Moisture Content and Silage Starter on the Nutritive Value of WRB Mixed Silage and WRPP Mixed Silage*

The wilting of high-MC silage raw materials before silage can promote fermentation and the long-term preservation of silage feed [38]. In this study, the DM and CP contents of WRB mixed silage and WRPP mixed silage in the 55% MC treatment group were significantly greater than those in the 60% MC and 65% MC treatment groups, indicating that the low-MC treatment did not result in nutrient loss, which agreed with the findings of Cavallarin et al. [39]. This is explained by the fact that a high MC increases the rate of nutrient loss with juice exudation and affects the nutritive value of silage, while a lower MC promotes former Lactobacillus fermentation and lowers the pH, thus inhibiting undesirable microbial activity, reducing the respiration of plant cells, and retaining more nutrients. On the other hand, the increased DM content in the 55% MC treatment group could effectively increase the DM intake of livestock during feeding, which positively affects livestock performance and breeding efficiency. Moreover, the NDF, ADF, and ADL contents of the WRB mixed silage and WRPP mixed silage decreased with a decreasing MC, probably because of the decreased DM loss during ensiling. Hashemzadeh-Cigari et al. [40] investigated the effect of wilting treatment on the nutritive value of lucerne silage and reported that the wilting treatment increased the DM content of the lucerne silage, decreased the NDF and ADF contents, and improved the quality of the silage, which agrees with the results of the present study. In general, the quality of wilted silage was better than that of unwilted silage [41].

WRB mixed silage and WRPP mixed silage are new silage products with high feeding value and economic benefits. The nutrient composition is an important indicator for determining the quality of silage. The key index for evaluating the efficiency of pasture utilisation is the DM content; the higher the DM content is, the greater the economic efficiency of pastures. In this experiment, the DM, OM, and CP contents of the WRPP mixed silage from the L group were greater than those in the control group at 55% MC, indicating that the addition of L preserved the nutrient content of the silage. The addition of former Lactobacillus strains can regulate the microbiota in the silage and accelerate the fermentation process, thus effectively preserving the nutrient content [42]. For example, Lactobacillus-treated silage increased the abundance of Lactobacillus and inhibited the activity of undesirable microorganisms (*Acinetobacter*) [43]. Previous studies have shown that the relative abundance of former Lactobacillus in fresh silage material is relatively low and increases gradually during the ensiling process [44]. Jiang et al. [45] showed that former Lactobacillus species enhanced the fermentation quality of maize silage by increasing the lactic acid concentration during the pre-fermentation period and lowering the pH, while former Lactobacillus species could regulate the microbiota and increase the relative abundance of former Lactobacillus species to better conserve silage nutrients, which agrees with the above experimental results. Microorganisms require energy for growth during silage fermentation, and WSCs are the main carbon source needed for the growth and reproduction of former Lactobacillus. In this study, RB and RPP were used as auxiliaries to promote the growth and reproduction of former Lactobacillus by increasing the WSC content, rapidly acidifying the silage, inhibiting the growth and reproduction of harmful microorganisms, and reducing the DM and CP losses. Chen et al. [46] inoculated alfalfa silage with former Lactobacillus and reported that the addition of former Lactobacillus promoted more effective homofermentation of the alfalfa silage, reduced the loss of DM and the decomposition of CP, and improved the nutritional value of the silage, which is consistent with the results of the present study. The NDF and ADF contents of the WRPP mixed silage from the L group were lower than those in the control group at 55% MC, which may be related to the degradation of the digestible components of the cell wall by the organic acids produced by the fermentation of former Lactobacillus [47]. The results of the above experiments showed that organic acids degrade the hemicellulose content of the cell wall, which agrees with the findings of Hristov et al. [48].

### 4.3. Effects of Moisture Content and Silage Starter on the Energy and In Vitro Digestibility of WRB Mixed Silage and WRPP Mixed Silage

Energy value is an important aspect of forage nutritional value assessment, and forage energy value assessment includes GE, DE, and ME assessments. Adequate energy intake helps to improve the performance of livestock and poultry and the economic efficiency of farmers. As previously mentioned, reducing the MC can retain more nutrients, which may be an important factor in the observed increase in DE and ME in this study. In this experiment, for the same L group, with a decreasing MC, the GE, DE, and ME of the WRB mixed silage gradually increased. This suggested that a low MC favoured energy accumulation in the silage. The level of DE is strongly influenced by the nutrient content of the forage; the higher the crude fibre content is, the lower the DE value. At 55% MC, the addition of L effectively enhanced the DE and ME of the WRPP mixed silage compared to those of the control. The addition of L improved the energy values of the WRPP mixed silage, possibly because L promoted the degradation of crude fibre in the silage.

In the same L subgroup, the IVDMD, IVOMD, and IVCPD of the WRB mixed silage in the 55% MC treatment group were significantly greater than those in the 65% MC treatment group, with increases of 4.9%, 4.7%, and 6.1%, respectively. This was in line with the findings of Wan et al. [11], who reported that low-MC treatment reduced the rate of protein hydrolysis in Sudan grass silage and that the IVDMD was significantly greater than that in the high-MC treatment. This was explained by the fact that the low MC treatment reduced the loss of DM and CP from the silage, better preserved the nutrient content of the silage, and improved the utilisation of the fermentation substrate in the rumen fermentation process, thus resulting in a higher IVDMD, IVOMD, and IVCPD. Cavallarin et al. [39] reported that protein hydrolysis in silage with a DM content less than 32% FW increased after butyric acid fermentation, which was consistent with the results of the present study. Previous studies have shown that the use of Lactobacillus as a silage inoculant may have a probiotic effect on rumen fermentation, specifically by maintaining the ecological balance of the flora while simultaneously interacting with rumen microorganisms to enhance the rumen function and animal performance [49]. Feeding silage inoculated with homofermentative former Lactobacillus preparations increased the milk production, daily weight gain, and feed efficiency in dairy cows [50]. Moran et al. [51] investigated the effect of *Lactiplantibacillus plantarum*-inoculated silage on the performance of beef cattle and showed that the dry matter intake and body weight gain of beef cattle fed *Lactiplantibacillus plantarum*-inoculated silage increased by 7.5 and 11.1%, respectively, compared to those of the control group. In this experiment, the IVDMD and IVCPD were greater for the WRB mixed silage and WRPP mixed silage in the L group than for the control at 55% MC. This may be because L improves the silage fermentation quality, inhibits undesirable microbial fermentation, and reduces DM and CP losses, thus increasing the IVDMD and IVCPD. According to Peltekova et al. [52], the addition of former Lactobacillus resulted in an increase in the IVCPD of silage, mainly due to the improved synchronisation between the rate of nitrogen release and the rate of energy fermentation resulting from an increase in the concentration of true protein. Aksu et al. [49] showed that inoculation with former Lactobacillus increased the IVDMD, IVCPD, and IVNDFD of maize silage. The inoculation of alfalfa silage with former Lactobacillus increased the IVDMD and IVCPD of the silage, which agreed with the results of this experiment [46]. NDF and ADF contents are important indicators for evaluating the balance of the animal feed concentrate/crude ratio and can indirectly affect the digestibility of ruminants by influencing their salivary secretion and chewing time. The lower the NDF and ADF contents are, the greater the nutrients available for absorption and utilisation by livestock and the greater the feeding value. The IVNDFD of the WRB mixed silage and WRPP mixed silage in the L group was greater than that in the control group at 55% MC, probably due to the degradation of the fibre content in the cell wall by the organic acids produced by the fermentation of former Lactobacillus. According to Table 1, RB had higher OM and CP contents and lower fibre content than RPP. This difference may explain why the in vitro digestibility of the

WRB mixed silage was greater than that of the WRPP mixed silage. As shown in Table 2, the fermentation quality of the WRB mixed silage increased gradually with a decreasing MC. If high-moisture-content WRB mixed silage inoculated with homofermentative former Lactobacillus are fed to ruminants, the growth performance of the ruminants may be affected by poor fermentation quality. Keady et al. [53] reported that silage inoculated with former Lactobacillus improved the animal performance. Khuntia et al. [54] determined the nutrient digestibility of maize silage with sheep and showed that silage supplemented with former Lactobacillus increased the DM intake, IVDMD, IVCPD, and IVNDFD in sheep. We can expect that WRB mixed silage inoculated with L will promote the ruminant dietary nutritional balance and improve the ruminant performance if it is introduced into ruminant diets at 55% MC.

## 5. Conclusions

Based on the results of this study, it is recommended to simultaneously reduce the MC and add L to WRB mixed silage and WRPP mixed silage to enhance the fermentation quality, nutritional value, and in vitro digestibility of both silage types. However, in terms of in vitro digestibility, WRB mixed silage improved better than WRPP mixed silage. Overall, the best results were obtained with the addition of L to WRB mixed silage and WRPP mixed silage at 55% MC of the silage material. In future studies, feeding trials are needed to assess the effect of L-treated WRB mixed silage (55% MC) on the growth performance of ruminants.

**Author Contributions:** Conceptualisation, Q.Y., B.Y. and P.W.; methodology, Q.Y., M.Y., J.D., Y.J. and H.T.; software, Q.Y., T.Z., M.Y., J.D. and Y.J.; validation, B.Y., P.W., J.D., Y.J., M.Y. and T.Z.; formal analysis, Q.Y., P.W. and J.D.; investigation, M.Y.; resources, B.Y. and H.T.; data curation, Q.Y., P.W. and M.Y.; writing—original draft preparation, Q.Y. and P.W.; writing—review and editing, Q.Y., P.W. and T.Z.; visualisation, B.Y. and P.W.; supervision, B.Y. and P.W.; project administration, B.Y. and P.W.; funding acquisition, P.W. and Y.J. All authors have read and agreed to the published version of the manuscript.

**Funding:** This study was supported by funds from the China Agriculture Research System (CARS-37) and the Major Science and Technology Special Fund for the Development of Beef Cattle Industry in Jilin Province (YDZJ202203CGZH042).

**Institutional Review Board Statement:** This study strictly followed the Chinese Laboratory Animal Welfare Ethical Review Guidelines and was approved by the Laboratory Animal Welfare Ethics Committee of Jilin University (permit number SY202009600).

**Informed Consent Statement:** Not applicable.

**Data Availability Statement:** The authors are solely responsible for the completeness and accuracy of all data in the article. Further inquiries can be directed to the corresponding authors.

**Acknowledgments:** We thank Zhihui Zhang and Tongyu Xinwei at Beixian Rice Co. for providing the test material.

**Conflicts of Interest:** The authors declare no conflict of interest.

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
