# Peer review of "Effects of Moisture Content and Silage Starter on the Fermentation Quality and In Vitro Digestibility of Waxy Corn Processing Byproduct Silage"

_fermentation, doi:10.3390/fermentation9121025_

Round 1

Reviewer 1 Report

Comments and Suggestions for Authors

Which were the process variable? Why? Include in text.

In the abstract text there are many “lower” and “higher”. This is not a report of results is a research article. Rewrite text to show your results and why change the properties.

In the manuscript author use a lot the words “lower” and higher”, my recommendation is that author avoid to write that 2 is lower than 5.

Line 224. Include a paragraph on the meaning of chemical properties in mixtures. For example, why RB has the highest crude protein content. Avoid use the word low, high, 2 times more, etc.

Line 225. Add paragraph as conclusion of the chemical composition.

Line 230. Explain i9n  text what is the “ fermentation quality” and how is measured and interpreted. Avoid use “lower” and “higher”. Better discuss differences and the effect of the differences.

Line 255. Add paragraph as conclusion of the fermentation quality.

Line 256. What is the difference between chemical composition in this line 256 and the chemical composition of the line 212. Explain in text.

Line 255. Add paragraph as conclusion of the chemical composition

Line 301. Add paragraph discussing the importance of this energy measurements. 

Line 302, Add paragraph as conclusion of the energy.

Line 325. Add paragraph as conclusion of the invitro digestibility.

Author Response

Dear reviewer, thank you for your encouraging and warm comments and suggestions, all of your suggestions are very important, and they all have important guiding significance for our future research work. Based on this we have revised and (we think) strengthened our paper.

Point 1: Which were the process variable? Why? Include in text.

Response 1: We fully appreciate your suggestion. The process variables were moisture content and additives. In this experiment, a two-factor (moisture content × additives) completely randomised experimental design was used to adjust the moisture content of the silage material to 55%, 60%, and 65% with rice bran and rice polished powder, using the waxy corn processing byproduct as silage material. At each moisture content condition, the silage treatments included groups without any additives (control) and with lactic acid bacteria (L). In the paper we add that section. (PDF:18-23)

Point 2: In the abstract text there are many “lower” and “higher”. This is not a report of results is a research article. Rewrite text to show your results and why change the properties.

Response 2: We fully appreciate your suggestion. We revised that section in the paper. (PDF:26-36)

Point 3: In the manuscript author use a lot the words “lower” and higher”, my recommendation is that author avoid to write that 2 is lower than 5.

Response 3: We fully appreciate your suggestion. We avoided the use of "lower" and "higher" in the paper. (PDF:13,206)

Point 4: Line 224. Include a paragraph on the meaning of chemical properties in mixtures. For example, why RB has the highest crude protein content. Avoid use the word low, high, 2 times more, etc.

Response 4: We fully appreciate your suggestion. The highest crude protein content of rice bran may be due to the fact that more endosperm is mixed with the rice bran raw material and its crude protein content is also higher. In the paper we avoided the use of words such as "low", "high" and "two times higher".

Point 5: Line 225. Add paragraph as conclusion of the chemical composition.

Response 5: We fully appreciate your suggestion. We revised that section in the paper. (PDF:224-227)

Point 6: Line 230. Explain i9n  text what is the “ fermentation quality” and how is measured and interpreted. Avoid use “lower” and “higher”. Better discuss differences and the effect of the differences.

Response 6: We fully appreciate your suggestion. Fermentation quality is one of the criteria for assessing the quality of silage, and is mainly characterised by chemical analyses, determining pH, the ratio of ammoniacal nitrogen to total nitrogen, and the content of organic acids. In section 2.2 of the article, we describe in detail the methods for determining the indicators related to fermentation quality. In the paper we avoid the use of 'lower' and 'higher'. (PDF:139-152,233-240)

Point 7: Line 255. Add paragraph as conclusion of the fermentation quality.

Response 7: We fully appreciate your suggestion. In the paper we add that section. (PDF:247-249)

Point 8: Line 256. What is the difference between chemical composition in this line 256 and the chemical composition of the line 212. Explain in text.

Response 8: We fully appreciate your suggestion. In line 212, it is the chemical composition of three silage ingredients (waxy corn processing byproduct, rice bran, and rice polished powder). In line256, it is the chemical composition of two mixed silages (waxy corn processing byproduct and rice bran, waxy corn processing byproduct and rice polished powder).

Point 9: Line 275. Add paragraph as conclusion of the chemical composition

Response 9: We fully appreciate your suggestion. In the paper we add that section. (PDF:273-275)

Point 10: Line 301. Add paragraph discussing the importance of this energy measurements. 

Response 10: We fully appreciate your suggestion. In the paper we add that section. (PDF:483-495)

Point 11: Line 302, Add paragraph as conclusion of the energy.

Response 11: We fully appreciate your suggestion. In the paper we add that section. (PDF:293-295)

Point 12: Line 325. Add paragraph as conclusion of the invitro digestibility.

Response 12: We fully appreciate your suggestion. In the paper we add that section. (PDF:315-317)

Thank you again for your suggestions and hope to learn more from you.

Reviewer 2 Report

Comments and Suggestions for Authors

This study showed a good silage-making formula to improve the silage quality and feeding value. However, the introduction section could be more brief.

But there are some significant revisions to be made, as outlined in the following summary of comments: 

L128

please define the "FW". is it means "fresh weight"?

L139-142

How to release the gas that is produced during the ensilage process in this study?

L161

Is there any aerobic stability assay after opening the silage product in this study?

L191

How about the type of artificial rumen incubator? The ratio of sample to incubation fluid will affect the digestion performance.

L192

How about the pH before and after incubation? The abnormal pH during the incubation process may result in incorrect digestibility.

L256

The data in Table 3 showed the silage product after 60 days ensilage. However, the chemical composition of the silage at day 0 should be shown in Table 1. 

L269-274

The DM losses after silage should be calculated to evaluate the energy conversion efficiency. The lactic acid bacteria supplementation could increase the lactic acid product, but it also increases the carbohydrate utilization during the fermentation process.

L284-293

The energy calculation results without the data at day 0 result in the difficulty of judging the optimal silage-making formula.

L327-328

The energy assay data of silages in Table 3 was well, but no discussion about energy data was shown in this manuscript.

L368-369

The data in Table 2 is from the silage product at 60 days, the degree of protein degradation in the silage should be calculated according to the difference before and after ensilage.

L410-413

The pH and VFA data after different ensilage days were not shown in this study, the present data may not be enough to explain the fermentation result.

L436-438

Compared to the high moisture silage, the decrease of NDF, ADF, and ADL in the lower moisture silage may result from less DM losses during ensilage. 

L442-448

The waxy corn is a Poaceae plant, discussion about the grass silage data to explain the present study result is more appropriate.

L477-480

The discussion in this section is very similar to L410-413.

L490-495

To discuss the benefit of inoculant bacteria in rumen fermentation, the author should show that the lactic acid bacteria that were added in silage making was still alive in the silage product. 

L506-527

The feeding value and chemical composition of silages in this study are recommended to compare with the normal corn silage with or without inoculation.

L529-531

The data in the Tables implied that only decreasing the moisture had improved the silage quality significantly. However, the digestibility of the WRPP treatment seems lower than WRB treatment, so the author should discuss the results of this study. 

The DM losses and aerobic stability data need to be considered when selecting the best silage-making condition. 

Author Response

Dear reviewer, thank you for your encouraging and warm comments and suggestions, all of your suggestions are very important, and they all have important guiding significance for our future research work. Based on this we have revised and (we think) strengthened our paper.

Point 1: L128 please define the "FW". is it means "fresh weight"?

Response 1: We fully appreciate your suggestion. FW is the fresh weight. We revised that section in the paper. (PDF:124)

Point 2: L139-142 How to release the gas that is produced during the ensilage process in this study?

Response 2: We fully appreciate your suggestion. The polythene vacuum bags we use have a one-way valve that allows air to be vented only outwards and not inwards. Thus, gases produced during the silage process can be released.

Point 3: L161 Is there any aerobic stability assay after opening the silage product in this study?

Response 3: We fully appreciate your suggestion. In this study, we did not perform aerobic stability measurements after opening the silage. Thanks again for the reminder that aerobic stability data needs to be considered when choosing optimal silage making conditions in the future.

Point 4: L191 How about the type of artificial rumen incubator? The ratio of sample to incubation fluid will affect the digestion performance.

Response 4: We fully appreciate your suggestion. In this study, the artificial rumen incubator we used was an electrically heated thermostatic water bath, type  (BZ-SHH-W21, Biaozhuo Scientific Instruments Co. Ltd., Shanghai, China).

Point 5: L192 How about the pH before and after incubation? The abnormal pH during the incubation process may result in incorrect digestibility.

Response 5: We fully appreciate your suggestion. In this study, pH values before and after incubation were not determined. The abnormal pH during the incubation process may result in incorrect digestibility. In future studies, we will first consider the determination of pH before and after incubation.

Point 6: L256  The data in Table 3 showed the silage product after 60 days ensilage. However, the chemical composition of the silage at day 0 should be shown in Table 1.

Response 6: We fully appreciate your suggestion. In this study, we followed the experimental design of Mu et al. (2020) (https://doi.org/10.1016/j.biortech.2020.123772) and carried out the determination of the chemical composition of the three silage ingredients (waxy corn processing byproduct, rice bran, and rice polished powder) and did not carry out the determination of the chemical composition of the day 0 mixed silage. Thanks again for the reminder that it is important to consider chemical composition determination of day 0 mixed silage.

Point 7: L269-274 The DM losses after silage should be calculated to evaluate the energy conversion efficiency. The lactic acid bacteria supplementation could increase the lactic acid product, but it also increases the carbohydrate utilization during the fermentation process.

Response 7: We fully appreciate your suggestion. The DM losses after silage should be calculated to evaluate the energy conversion efficiency. In this study, we did not carry out chemical composition measurements of day 0 mixed silage to the extent that DM loss rates could not be calculated at this time. Thanks again for the reminder that it is important to consider chemical composition determination of day 0 mixed silage.

Point 8: L284-293 The energy calculation results without the data at day 0 result in the difficulty of judging the optimal silage-making formula.

Response 8: We fully appreciate your suggestion. The energy calculation results without the data at day 0 result in the difficulty of judging the optimal silage-making formula. In this study, we followed the experimental design of Mu et al. (2020) (https://doi.org/10.1016/j.biortech.2020.123772) and carried out the determination of the energy of the three silage ingredients (waxy corn processing byproduct, rice bran, and rice polished powder) and did not carry out the determination of the energy of the day 0 mixed silage. Thanks again for the reminder that it is important to consider energy determination of day 0 mixed silage.

Point 9: L327-328 The energy assay data of silages in Table 3 was well, but no discussion about energy data was shown in this manuscript.

Response 9: We fully appreciate your suggestion. We add a discussion of energy in Article 4.3. (PDF:483-495)

Point 10: L368-369 The data in Table 2 is from the silage product at 60 days, the degree of protein degradation in the silage should be calculated according to the difference before and after ensilage.

Response 10: We fully appreciate your suggestion. The lower the degree of CP degradation is, the smaller the AN/TN is. However, the degree of CP degradation in silage should be calculated based on the difference between before and after silage. In this study, we did not perform chemical composition measurements of day 0 mixed silage to the extent that CP degradation rates could not be calculated at this time. Thanks again for the reminder that it is important to consider chemical composition determination of day 0 mixed silage.

Point 11: L410-413 The pH and VFA data after different ensilage days were not shown in this study, the present data may not be enough to explain the fermentation result.

Response 11: We fully appreciate your suggestion. The pH and VFA after different days of silage were not measured in this trial. Thanks again for the reminder that in future studies we will first consider determining pH and VFA after different days of silage.

Point 12: L436-438 Compared to the high moisture silage, the decrease of NDF, ADF, and ADL in the lower moisture silage may result from less DM losses during ensilage. 

Response 12: We fully appreciate your suggestion. We revised that section in the paper. (PDF:438-439)

Point 13: L442-448 The waxy corn is a Poaceae plant, discussion about the grass silage data to explain the present study result is more appropriate.

Response 13: We fully appreciate your suggestion. We revised that section in the paper. (PDF:498-500)

Point 14: L477-480 The discussion in this section is very similar to L410-413.

Response 14: We fully appreciate your suggestion. In the paper, we have deleted L477-480. (PDF:467-471)

Point 15: L490-495 To discuss the benefit of inoculant bacteria in rumen fermentation, the author should show that the lactic acid bacteria that were added in silage making was still alive in the silage product.

Response 15: We fully appreciate your suggestion. In order to discuss the benefits of inoculated bacteria in rumen fermentation, it should be demonstrated that lactic acid bacteria added during silage making are still alive in the silage product. From Table 2, after 60 days of silage, the pH of all treatment groups was much less than 4.2, indicating good fermentation, and in another way proving that the lactic acid bacteria added during the silage process are still alive in the silage product. Thank you again for the reminder that in future studies we will use the plate trace surface drop seeding method to test the number of viable lactic acid bacteria in silage that has been supplemented with Lactobacillus solution during the silage process to demonstrate if they are still viable.

Point 16: L506-527 The feeding value and chemical composition of silages in this study are recommended to compare with the normal corn silage with or without inoculation.

Response 16: We fully appreciate your suggestion. In this study, we did not perform inoculated or uninoculated common corn silage preparation. Thanks again for the reminder that in future studies we will first consider carrying out inoculated or uninoculated common corn silage preparations to the point of comparing them with mixed silage.

Point 17: L529-531 The data in the Tables implied that only decreasing the moisture had improved the silage quality significantly. However, the digestibility of the WRPP treatment seems lower than WRB treatment, so the author should discuss the results of this study.

Response 17: We fully appreciate your suggestion. In the paper we add a discussion of this result. (PDF:535-558)

Thank you again for your suggestions and hope to learn more from you.

Round 2

Reviewer 2 Report

Comments and Suggestions for Authors

1. Each bath of corn plant may show a different chemical composition, the corn data from the previous study is not suitable to be applied in different studies.

2. The author could calculate the in vitro true dry matter digestibility (IVTDMD) according to the NDF and DM digestibility. It may present a more comprehensive result.

Author Response

Dear reviewer, thank you for your encouraging and warm comments and suggestions, all of your suggestions are very important, and they all have important guiding significance for our future research work. Based on this we have revised and (we think) strengthened our paper.

Point 1: Each bath of corn plant may show a different chemical composition, the corn data from the previous study is not suitable to be applied in different studies.

Response 1: We fully appreciate your suggestion that each bath of corn plant may show a different chemical composition. In this trial, we did not use maize data from previous studies. Table 1, it is the chemical composition of three silage ingredients (waxy corn processing byproduct, rice bran, and rice polished powder). Table 3, it is the chemical composition of two mixed silages (waxy corn processing byproduct and rice bran, waxy corn processing byproduct and rice polished powder). The data for this experiment were all for the first use.

Point 2: The author could calculate the in vitro true dry matter digestibility (IVTDMD) according to the NDF and DM digestibility. It may present a more comprehensive result.

Response 2: We fully appreciate your suggestion. We could calculate the in vitro true dry matter digestibility (IVTDMD) according to the NDF and DM digestibility. It may present a more comprehensive result. After reviewing the literature, we were unable to find a relevant formula for how to use NDF and DM digestibility to calculate IVTDMD, and in previous studies, IVTDMD has never been addressed by our group. Therefore, we sincerely hope that the teacher can tell us how to use NDF and DM digestibility to calculate IVTDMD, so that we can use this formula in this study and future research. Thank you very much, teacher! Email: 13540649853@163.com

Thank you again for your suggestions and hope to learn more from you.
